# Cell-Free DNA Analysis within the Challenges of Thyroid Cancer Management

**DOI:** 10.3390/cancers14215370

**Published:** 2022-10-31

**Authors:** Vincenzo Marotta, Michele Cennamo, Evelina La Civita, Mario Vitale, Daniela Terracciano

**Affiliations:** 1UOC Clinica Endocrinologica e Diabetologica, AOU San Giovanni di Dio e Ruggi d’Aragona, 84131 Salerno, Italy; 2Department of Translational Medical Sciences, University of Naples “Federico II”, 80138 Naples, Italy; 3Dipartimento di Medicina, Chirurgia e Odontoiatria, Università di Salerno, 84081 Baronissi, Italy

**Keywords:** thyroid cancer, thyroid nodules, thyroid diseases, liquid biopsy, cell-free DNA

## Abstract

**Simple Summary:**

Liquid biopsy is a minimally invasive method that emerged as a new promising tool for improving diagnosis, risk stratification, follow-up, and treatment of cancer patients. To date, the majority of the research in the area of liquid biopsy has focused on plasma-based cell-free DNA as a potential surrogate for tumor DNA obtained from a tissue biopsy. In the last decades, breakthrough advancements have been performed in the knowledge of thyroid cancer genetics, and the role of molecular characterization in clinical decision-making is continuously rising, from diagnosis completion to the personalization of treatment approach. Hence, it is expectable for cell-free DNA to be applicable in thyroid cancer management. This review aims to investigate the cell-free DNA utility for thyroid cancer patients’ care.

**Abstract:**

Thyroid cancer is the most frequent endocrine malignancy with an increasing incidence trend during the past forty years and a concomitant rise in cancer-related mortality. The circulating cell-free DNA (cfDNA) analysis is a patient’s friendly and repeatable procedure allowing to obtain surrogate information about the genetics and epigenetics of the tumor. The aim of the present review was to address the suitability of cfDNA testing in different forms of thyroid cancer, and the potential clinical applications, as referred to the clinical weaknesses. Despite being limited by the absence of standardization and by reproducibility and validity issues, cfDNA assessment has great potential for the improvement of thyroid cancer management. cfDNA may support the pre-surgical definition of thyroid nodules by complementing invasive thyroid fine needle aspiration cytology. In addition, it may empower risk stratification and could be used as a biomarker for monitoring the post-surgical disease status, both during active surveillance and in the case of anti-tumor treatment.

## 1. Introduction

Thyroid cancer is the most common endocrine malignancy with an incidence of about 14 cases per 100,000 inhabitants/years in the United States [1]. Furthermore, in the last four decades, its incidence tripled [2]. Even more importantly, the latest thyroid cancer report by the US Surveillance, Epidemiology, and End Results showed a concomitant increase in cancer-related mortality, as occurred in the last two decades [2].

The vast majority of thyroid malignancies are classified as differentiated thyroid cancer (DTC) [3]. DTC includes a heterogeneous group of follicular-derived tumors: (a) papillary thyroid carcinoma (PTC) is the most common (about 85% of thyroid malignancies) and is characterized by excellent prognosis, with nearly 100% 5-year disease-specific survival [4]; (b) follicular thyroid cancer (FTC) accounts for about 5% of cases and shows poorer prognosis [5], due to higher tendency to locoregional invasion [6] and to metastatic spread [7]; (c) Hürthle cell carcinoma (HTC) represents about 3% of thyroid cancers [8] and is more aggressive, as compared with both PTC and FTC, with higher rates of distant metastases and reduced progression-free survival [9]; (d) poorly differentiated thyroid cancer (PDTC) shows variable prevalence, from 2 to 15% [10], and is highly aggressive, with a strong tendency to metastasize and a 5-year disease-specific survival of 66% [11]. In addition to the DTC *spectrum*, there is another type of follicular-derived thyroid malignancy, anaplastic thyroid cancer (ATA), accounting for less than 2% of malignant thyroid tumors. Its prognosis is abysmal, having less than 6 months mean survival [12]. Finally, there is one form of non-follicular thyroid malignancy, medullary thyroid cancer (MTC), originating from the parafollicular calcitonin (CT)-secreting C cells, which accounts for 2–3% of thyroid carcinomas. MTC is quite aggressive, with a reported cancer-specific mortality up to 38% at 10 years [13].

Despite the wide molecular characterization of the vast majority of thyroid malignancies and the effort to reduce such a body of knowledge into suitable tools for disease management [14], physicians dealing with thyroid cancer have still to reckon with several challenging issues, from diagnostics to therapeutics.

Cell-free DNA (cfDNA) consists of DNA fragments released by the cells into the bloodstream. Physiologically, an amount of cfDNA derives from the apoptosis of hematopoietic cells [15]. In cancer patients, there is an adjunctive portion of cfDNA generated by the necrosis of neoplastic cells [16], defined as circulating tumor DNA (ctDNA). ctDNA has its own properties (length, size, and stability) [17] and, more importantly, carries the genetic and epigenetic signatures of the tumor [18,19].

For many years, a wide research effort is ongoing with the aim of finding out possible clinical applications for cfDNA analysis in cancer management.

Up to now, the most promising use is for cancer genomics, defined as the detection of tumor-related genetic alterations, to be used for prognostic stratification, therapeutic interventions, and, more recently, for the detection of drug resistance [20,21]. Historically, analysis of tumor genetics was performed by tissue biopsy. In addition to being limited by anatomical accessibility and possible adverse events, tissue analysis cannot be repeated many times through the disease course, thus being unable to follow mutational status [22]. Furthermore, genetics from a single neoplastic site cannot be considered representative of the whole disease, especially in high-burden tumors. By contrast, blood-based cancer genomics (the so-called liquid biopsy) may be easily performed at different times, allowing a dynamic mutational assessment [23] able to capture clonal differences between different sites, obtaining a picture of the tumor heterogeneity [24]. In such context, analysis of cfDNA is very promising, due to the high concordance with genetic and epigenetic characteristics of tumor tissue [25,26], coupled with a reduced turnaround time [27].

In addition to the assessment of tumor genetics, the evaluation of cfDNA quantitative and qualitative features, particularly the concentration [28] and the fragmentation pattern [29], have been studied as markers of tumor burden and morphological slope (spontaneous or upon antitumor treatments), and as predictors of clinical outcome.

The present review was aimed at reporting available evidence about cfDNA analysis in thyroid cancer patients. Particularly, applied detection methods and their accuracy were analyzed and the applications within the various clinical controversies were addressed.

## 2. Methods

A literature search was performed on MEDLINE. We used as search terms and Boolean operators: “cell free DNA” OR “circulating DNA” OR “circulating cell free DNA” OR “cell free nucleic acids” OR “circulating mutations” OR “liquid biopsy” AND “thyroid cancer” OR “thyroid neoplasms” OR “papillary thyroid” OR “follicular thyroid” OR “poorly differentiated thyroid” OR “medullary thyroid” OR “anaplastic thyroid”. We set filters to select human studies published in the last 15 years. No language restrictions were applied.

All thyroid cancer histologies were considered. Both prospective and retrospective designs were included. Studies with less than 20 patients were excluded.

Data extraction included the following items: study design; thyroid cancer histology; patients’ clinico-pathological features; applied techniques and biological matrix; type of cfDNA analysis; main results including metrics parameters of the cfDNA testing.

Due to the heterogeneity of study designs, collected data, and outcome assessment, a statistical meta-analysis of included studies was not feasible. Therefore, results were presented in a narrative form.

## 3. cfDNA Analysis in Thyroid Cancers

Many approaches have been used for the detection and characterization of cfDNA in oncology, including thyroid cancer, with peculiar technical features, costs, turnaround time, and clinical performance.

Quantitative real-time PCR (qPCR) was the first to be applied, easy to use, and cheap, allowing amplification of nucleic acids with low turnaround time. The main limit of qPCR is the low accuracy for ctDNA characterization. Therefore, due to the introduction of more accurate techniques (see below), the optimal application of such an approach should be the analysis of cfDNA quantitative and qualitative features [30].

Recently, a new PCR-based method has been developed, named digital PCR (dPCR). This technique is based on the partition of the sample in many individual reactions using droplet-based or microfluidic platform systems and is strikingly sensitive for the detection of low levels of mutated DNA variants out of a number of wild type sequences [31]. Hence, in the context of cancer-related cfDNA analysis, it is accurate for the detection and characterization of genetic/epigenetic tumor features. dPCR is cheaper, as compared to the classical sequencing systems used for absolute quantification and mutation detection. The disadvantages are that known variants can be screened and only a few of them can be analyzed simultaneously [32].

The next-generation sequencing (NGS) technology is based on the parallel analysis of millions of short DNA sequences followed by alignment to a reference genome. This approach is highly sensitive allowing to detect mutations with less than 0.2% allele frequency and to sequence target regions with high coverage (10.000x) [33]. Therefore, NGS is the most effective method for performing a wide-throughput genetic analysis. In the context of cancer-related cfDNA analysis, the best NGS application is the comparative assessment of tissue and circulating DNA. Limitations are the high cost and complexity of the analysis and data interpretation.

The majority of studies analyzing cfDNA in thyroid cancer are still based on qPCR [34,35,36,37,38,39,40,41,42,43,44]. However, in recent years, many papers applied NGS [45,46,47,48] and the dPCR [42,46,49,50,51,52,53].

### 3.1. Detection of Circulating Cell-Free BRAF^V600E^ in PTC

The majority of cfDNA studies about thyroid cancer performed to date were focused on the BRAF^V600E^ oncogene, representing the most common driver mutation occurring in PTC [54].

Results from studies analyzing the prevalence of circulating cell-free BRAF^V600E^ (cfBRAF^V600E^) among PTC carrying the mutation at the somatic level are widely discrepant, with values ranging from 0 to 91.7% [36,37,39,41,42,49,50] (Table 1). Considering the similar design of all mentioned papers, where the somatic analysis was performed on primary tumors and the cfDNA assessment before surgery, this variable sensitivity has to be attributed to technical issues, such as the accuracy of assay reagents and the adequacy of pre-analytical steps. Of note, three studies [42,49,50] applied the dPCR technology, the best-fitting tool for the detection of the small portion of mutated ctDNA out of the total amount of cfDNA. Among them, the Condello paper found no cfDNA mutated BRAF by both qPCR and dPCR, so the occurrence of technical issues can be assumed. The Jensen and Sato studies reported prevalence values of 42.1 and 31%, respectively. These results are consistent, especially considering that in the Jensen study, the analytical sensitivity was increased by the co-amplification at lower denaturation temperature-based PCR (COLD-PCR), which was aimed at amplifying BRAF mutated alleles before dPCR. Of note, Pupilli et al. [36] and Patel et al. [37] found cfDNA BRAF^V600E^ in 30 and 40%, respectively, of BRAF-negative PTC. This discrepancy could be related to the fact that mutated BRAF infrequently occurs as a clonal event [55,56], so somatic analysis may have been performed on a portion of the tumor not carrying the mutation. Based on such a hypothesis, the cfDNA analysis may allow for overcoming the issue of PTC genetic heterogeneity [57]. However, another possible explanation is the concomitant presence of other undiagnosed tumors carrying BRAF^V600E^, such as melanoma and colorectal cancer [58].

### 3.2. Detection of Circulating Cell-Free M918T RET in MTC

The M918T RET mutation represents the most common tumor-related alteration in MTC [59]. However, only one cfDNA study specifically focusing on such alteration has been performed up to now.

Cote et al. [51] searched for the M918T RET mutation by means of droplet dPCR. The study cohort included patients already subjected to primary tumor removal with persistent biochemical/structural disease, and, in the majority of cases, somatic assessment of M918T RET was accomplished on metastatic sites. Based on the single patient with an M918T RET-negative tumor showing circulating mutated alleles, the authors considered 0.21% and 1.36 as positivity cut-offs, for the allelic frequency and copies per plasma milliliter, respectively. The test yielded positive in 32 and 40% of advanced MTC harboring M918T RET, by considering allelic frequency and copies per plasma milliliter, respectively.

### 3.3. Tissue-Based Multigene cfDNA Studies

In recent years, a series of thyroid cancer cfDNA studies were based on the tumor tissue NGS assessment, as accomplished through dedicated multi-panel platforms, aimed at picturing the somatic status and directing the cfDNA analysis, which was targeted towards the detected tissue mutations.

In 2018, Allin et al. [52] focused on a cohort of thyroid cancers covering the whole histology spectrum. Following the NGS tissue analysis, cfDNA assessment was performed by means of dPCR. Of note, the plasma analysis was accomplished after surgery, at multiple time points during follow-up. Authors found cfDNA positivity, consisting in the circulating detection of the somatically identified mutations, in a percentage as high as 67%.

Two studies [46,48] used the NGS technology for analyzing tumor tissue and plasma from ATC patients. If focusing on patients naïve for any treatment at the time of the dual NGS assessment, there was a high concordance between the somatic and the cfDNA mutational status. Particularly, for the BRAF^V600E^ mutation, concordance was 100 and 92.9%, for the Sandulache and the Qin paper, respectively.

In 2020, Lan et al. [45] performed a matched tumor-plasma NGS analysis of a PTC cohort. Out of 36 patients subjected to cfDNA assessment, all carrying tumor-related alterations on the primary tumor, the authors found concordant circulating mutations in only 19.4% of cases. Actually, this was not attributable to a real tissue-plasma discrepancy, but to the low cfDNA detection rate, which was 38.9% in the overall cohort and even 20% in non-metastatic PTC. However, out of 21 cfDNA mutations identified, 18 (85.7%) were consistently found in the primary tumor, demonstrating remarkable concordance.

Very recently, Ciampi et al. [47] performed a matched tissue-plasma NGS study of a cohort of sporadic MTC. The somatic assessment was performed on primary tumors and the cfDNA analysis was carried out preoperatively only in case a somatic mutation was identified. Based on the analysis of two healthy controls, an allelic frequency >0.4% was established as a positivity cut-off. cfDNA testing yielded positive, with the tumor-related mutation being detected at both the somatic and the plasma analysis in 15.4% of the cohort.

### 3.4. Assessment of cfDNA Quantity and Quality

Four studies analyzed cfDNA purely quantitative and qualitative parameters in thyroid cancers [34,35,43,44]. All of them relied on the qPCR amplification of two-length cfDNA fragments. Targets of the amplification were the ALU repeats for the Zane and Higazi papers, the Amyloid Precursor Protein (APP), and the β-actin genes for the Salvianti and Klimaite studies, respectively. The authors focused on the absolute amplicons concentrations and on the so-called integrity index, consisting of the high/low length fragment ratio. This is a qualitative parameter expressing cfDNA fragmentation and reflecting the portion of necrotic derivation out of the total cfDNA amount. Indeed, tumor-related necrosis releases into circulation high-size DNA fragments, whereas those produced by the physiological apoptosis of blood cells are uniformly shorter than 200 bps [60]. Therefore, the higher the integrity index, the higher the likelihood and the entity of a cancer-related necrotic process.

The mentioned NGS study by Lan et al. [45] focused on crude cfDNA detection, assessing the prognostic implications (see below).

Ultimately, there is one study assessing the cfDNA methylation status by means of methylation-specific qPCR [61]. Hu and colleagues [38] focused on a cluster of five genes (CALCA, CDH1, TIMP3, DAPK, and RARβ2) showing a significant methylation level in a pre-analytical set of thyroid cancer patients, analyzing the possible clinical implications (see below).

## 4. Diagnostics

### 4.1. Clinical Challenges

The main diagnostic challenge is to discriminate the small portion of malignant thyroid nodules from the wide number of benign lesions [62].

Ultrasonography (US) represents the first-line tool for assessing the nature of thyroid nodules [63], but the positive predictive values (PPV) of the most applied US-based systems are abundantly lower than 50% [64]. Conversely, in pediatric subjects, the US risk stratification suffers from poor sensitivity and a high missing malignancy rate [65].

Thus, invasive thyroid fine needle aspiration cytology (FNAC) is frequently needed to refine diagnosis. Despite being highly accurate [66], FNAC yields no definite result in 25 to 30% of cases, due to inadequate sampling and the technical impossibility of defining the tumor nature. The latter generates the so-called indeterminate cytology [67], which represents the major challenge for thyroid cytopathology.

In the case of indeterminate results, molecular analysis of nucleic acids obtained from the FNAC sample may help to refine diagnosis, but its actual value is still a matter of debate [68].

### 4.2. Role of cfDNA Analysis

Characteristics and results of studies assessing the diagnostic role of cfDNA in thyroid cancer are reported in Table 2.

Overall, cfDNA-based analytical approaches were revealed as suitable for the pre-surgical identification of malignant nodules.

Two papers reported data about the diagnostic performance of circulating BRAF^V600E^.

In 2013, Pupilli et al. [36] showed that the percentage of cfBRAF^V600E^, as assessed by means of allele-specific qPCR, was significantly higher in PTC, as compared with benign goiter. However, the diagnostic accuracy in the Thy3 cytology was poor, with a 33% PPV. This was likely due to the low prevalence of the BRAF mutation in the malignancies related to the indeterminate cytology, where RAS-like thyroid cancers are the most common [70,71].

Recently, in 2021, the qPCR analysis by Patel et al. [37] found cfBRAF^V600E^ in 15 out of 109 thyroid nodule patients subjected to surgery (13.8%), all receiving a histological diagnosis of classic PTC. Therefore, in this series, circulating BRAF^V600E^ was able to predict malignancy with 100% specificity/PPV. Unfortunately, no data specifically referring to the indeterminate cytology were available.

The diagnostic significance of cfDNA amount and fragmentation was assessed in four papers.

Zane et al. [34] found that levels of both the high (ALU244) and the low (ALU83) length qPCR amplicon were higher in thyroid cancer, as compared with the healthy controls. ROC analysis identified cut-offs guaranteeing specificity of 100 and 94.7%, respectively. However, the lack of a subgroup of subjects with benign goiter did not allow a proper evaluation of the diagnostic performance.

The technically similar study by H”gazi’et al. [44] found that concentrations of the qPCR amplicons and the integrity index were higher in thyroid malignancies as compared not only with healthy subjects but also with benign goiter. Of note, in the indeterminate Bethesda IV category, specificity for thyroid cancer diagnosis was 100% for the integrity index and 91% for ALU -244 and -83 circulating amounts, respectively.

In 2021, Dutta et al. found that the total cfDNA concentration, as determined by means of biospectrometer, was higher in malignant nodules as compared with benign ones. The authors tested diagnostic accuracy in the indeterminate cytology (Bethesda III/IV categories) reporting sensitivity and specificity of 100 and 92.3%, respectively.

More recently, Klimaite et al. [43] found that the amount of both the high (β-actin394) and the low (β-actin99) length qPCR amplicons was higher in PTC, as compared with the healthy controls, whereas the integrity index (β-actin394/99 ratio) was higher as compared to both healthy patients and benign goiter. At ROC analysis, the AUC of the integrity index for differentiating PTC from benign goiter was 0.629, with sensitivity and specificity of 69.1 and 66.7%, respectively.

Ultimately, the mentioned Hu study [38], assessing the cfDNA methylation of five target genes (see above) prior to thyroid removal, was able to distinguish between DTC from benign goiter with high accuracy. The authors showed relevant specificity for each of the analyzed genes, ranging from 95 to 100%. Concerning the overall diagnostic performance of the test, specificity related to the positivity of at least one gene was 96% in the overall study population and 100% if focusing on the subgroup of patients with indeterminate cytology.

## 5. Prognostics

### 5.1. Clinical Challenges

Despite having a favorable prognosis, about 25–30% of DTC experience disease morbidity, as related to persistent structural disease/recurrence [72], and 10% die as related to cancer [73]. Identifying the DTC subgroup with the worst outcome represents the prognostic goal. However, available prognostic systems allow suboptimal long-term stratification, due to the low proportion of variance explained [74] and, more importantly, to the low PPV [75,76].

As compared with DTC, MTC is more aggressive and frequently persists upon initial surgery. Therefore, the prognostic goal is to predict survival. However, a dedicated prognostic system is missing, with the current TNM classification deriving from the DTC-related evidence and being unable to predict disease-related death [77].

### 5.2. Role of cfDNA

Characteristics and results of studies assessing the prognostic value of cfDNA in thyroid cancer are reported in Table 3.

The most robust evidence is about the prognostic significance of circulating tumor-related mutations.

In DTC, no cfDNA studies have assessed the association with long-term outcomes. However, there is evidence, as derived from four papers about the relationship between cfDNA BRAF^V600E^ and aggressive clinico-pathological factors.

The first one, by Kim et al. [39], found that circulating BRAF^V600E^, as assessed by means of qPCR was strongly related to the presence of distant metastases (specifically to the lung), as the test yielded positive in three out of four cases (75 and 100% as sensitivity and specificity, respectively).

More robust evidence was provided by the mentioned dPCR Jensen study [49], which reported a significant relationship between circulating BRAF^V600E^ and many features of disease aggressiveness in a cohort of BRAF-mutated PTC: high tumor size, gross extra-thyroidal extension, pulmonary micro-metastases, and high-risk American Thyroid Association (ATA) category [78]. However, the paper also demonstrated a strong correlation with the response to primary treatment (surgery ± radioactive iodine (RAI)), which is nowadays considered the most robust parameter for DTC prognostic stratification [74]. Authors found that cfBRAF^V600E^ independently predicted the non-excellent treatment response, consisting of the evident/suspected persistence of biochemical and/or structural disease [79], with an OR of 4.68. Of note, such a relationship remained significant if focusing on low-risk PTC, accounting for more than half of new diagnoses [80] and requiring feasible tools for the identification of the small portion of recurring patients. In such a subgroup, response to treatment was excellent just in one out of four (25%) cfDNA BRAF-positive patients and in 14 out of 15 (93%) cases showing the circulating wild type gene form. If looking at the positive predictive power of the test, the specificity and PPV of cfBRAF^V600E^ were remarkable, namely 93.3 and 80.4%, respectively.

As compared with the Jensen report, in the cited dPCR Sato paper [50], focusing on 16 BRAF-positive tumors, the correlation of cfBRAF^V600E^ with unfavorable clinico-pathological features was not as robust. Indeed, a significant relationship was found only with the extra-thyroidal extension. This could be attributed to the inclusion of patients with non-advanced disease, with any PTC carrying distant metastases and all cases classified as stage I/II (according to the AJCC TNM 8th Edition [81]). However, a relevant result was that PTC with detectable circulating BRAF^V600E^ showed a higher percentage of BRAF-mutated alleles in the primary tumor tissue. This finding may have notable prognostic implications, as a relationship between the fraction of mutated alleles and the poor outcome has been demonstrated [82].

Differently from the mentioned papers, the analysis by Patel et al. [37] included BRAF -positive and -negative tumors. The authors found that cfBRAF^V600E^ detection was related to a higher TNM stage and to extra-thyroidal extension.

Concerning MTC, there is evidence, as obtained by the dPCR Cote paper [51], of a direct prognostic impact for the cfDNA RET M918T mutation, which, when detected at the somatic level, is related to higher rates of metastasis and death [59]. Indeed, the plasma detection of such mutation in subjects with persisting post-surgical disease was related not only to the presence of distant metastasis but also to worse survival. Even circulating RET M918T demonstrated higher accuracy for outcome prediction, as compared with the CT doubling time, which is a recognized survival predictor [83].

As opposed to the Cote paper, the multi-gene cfDNA study by Ciampi et al. [47] performed the analysis preoperatively and in the immediate post-surgical phase (after 8 months median time). The detection of circulating tumor-related mutations was related to a higher risk of persistent disease, with all patients testing positive pre- or post-operatively presenting biochemical or structural disease upon surgery. Furthermore, the pre-operative cfDNA positivity was related to higher levels of the tumor-specific markers Ct and carcinoembryonic antigen (CEA), which have recognized prognostic impact [83,84].

Concerning ATC, the only available evidence as derived by the mentioned Qin NGS paper [48] is about the relationship between mutated cfPIK3CA and poor survival. Particularly, the survival of cfPIK3CA mutated and wild type ATC was 9.57 and 14.83 months, respectively.

Data about the prognostic significance of purely cfDNA-related parameters are more limited.

The mentioned study by Zane et al. [34] showed that concentrations of both the low (APP67) and high (APP180) length fragment as well as the integrity index (APP180/67 ratio) progressively increase across the thyroid cancer aggressiveness histology *spectrum*.

The NGS Lan study [45] found that cfDNA detection was related to distant metastases, higher tumor size, and invasiveness.

The Dutta paper [69] found that the total cfDNA amount positively correlated to lymph node metastasis, lymphovascular and capsular invasion, extra-thyroidal extension, and pTNM.

Ultimately, the Klimaite study [43] focused on PTC and reported significantly increased concentrations of the short-length cfDNA fragment β-actin99 and higher cfDNA integrity index β-(actin394/99) for tumors with >2 cm diameter.

## 6. Follow-Up Phase

### 6.1. Clinical Challenges

Due to the fact that about two-thirds of patients achieve structural cure following thyroid ablation [72], the main goal for DTC follow-up is to identify and perform the risk weight of recurrences.

Currently, DTC monitoring is based on the dynamic assessment of disease status at each follow-up visit [85].

The historical serum DTC marker, the thyroid-tissue specific glycoprotein thyroglobulin (Tg), is the most sensitive tool for disease detection upon complete thyroid ablation [86]. However, such diagnostic performance strictly depends on the concomitant presence of Tg antibodies (AbTg), reported in up to 25% of DTC, which may lead to underestimation of Tg levels and even to false negative results [87]. Furthermore, the accuracy of Tg may be impaired in aggressive DTC forms, characterized by a relevant grade of dedifferentiation, such as PDTC [88].

Another limitation is represented by the actual significance of the low-level Tg positivity, occurring in about 20% of patients subjected to RAI and, by definition, in subjects treated with surgery alone [89]. Indeed, PPV related to such weak biochemical evidence is low and there are no suitable cut-offs to be applied [90]. In this setting, it is crucial to observe the Tg slope over time [91], but prolonged observation is necessary and there is reduced accuracy.

As opposed to DTC, about two-thirds of MTC does not achieve disease eradication after surgery [92]. Therefore, most frequently, the follow-up objective is not to discover recidivisms in cured subjects through a time-by-time analysis of disease status, but to predict aggressive disease and poor outcomes in order to drive the surveillance approach. Owing to this, the MTC follow-up phase actually overlaps with the prognostics, which has been discussed in the previous paragraph.

### 6.2. Role of cfDNA

Up to now, the most pertinent data have been provided by two studies [38,40], which accomplished the cfDNA assessment in the post-surgical phase, with the aim of verifying whether the analyzed circulating parameter reflected the disease status.

The 2009 paper by Cradic et al. [40] searched for circulating BRAF^V600E^ in a cohort of 173 PTC by means of allele-specific qPCR. cfDNA mutated BRAF was found in 11.6% of cases and was usually consistent with the somatic status, with only one positive case not revealing the mutation on tumor tissue. The authors found that cfBRAF^V600E^ was related to a significantly higher (2.55 RR) likelihood of structural/biochemical disease. Particularly, test specificity was 91.7%, even though due to the low prevalence of non-disease-free patients, the PPV was only 44.6%.

The cfDNA 5-genes methylation test proposed by Hu et al. [38] (see above) yielded positive (at least one highly methylated gene) in 7 out of 10 (70%) recurring DTC, whereas the test was negative in 23 out of 29 (79.3%) disease-free cases. Overall, specificity was high, namely 88.5% even though PPV was only 61.6%. Furthermore, out of three false negative patients, just two showed weak biochemical evidence of disease.

## 7. Management of Advanced Disease

### 7.1. Clinical Challenges

Five percent of DTC develop metastatic RAI-refractory disease. The long-term survival of these patients is 10% [93].

Fifty-five percent of MTC experience metastatic disease. After metastases detection, survival is poor, 10% at 10 years [94].

To date, four KIs have been approved for these settings: sorafenib (for iodine-refractory DTC), lenvatinib (for iodine-refractory DTC), vandetanib (for MTC), cabozantinib (for MTC and as a second line for iodine-refractory DTC) [95,96].

However, the management of advanced DTC and MTC has several critical points.

Concerning iodine-refractory DTC, the utility of the tumor-specific marker Tg for clinical management is limited, as these tumors are characterized by a variable degree of dedifferentiation with decreased expression of the thyroid handling genes [97], including Tg. However, in the post-surgical management of DTC with a persistent structural disease, Tg did not show a linear correlation either with tumor burden and morphological slope [98]. Ultimately, ad-hoc biomarkers analyses from the sorafenib [99] and lenvatinib [100] phase III trials consistently showed that baseline Tg was not able to predict progression-free survival (PFS).

Concerning advanced MTC, the tumor-specific markers Ct and CEA, specifically their doubling time, are highly predictive of both survival and disease progression [83,84], and, due to the absence of antibody interference issues, may be considered as more reliable as compared with Tg. However, post-hoc analysis from the phase III trials of KIs in MTC [101,102] assessing Ct and CEA for the prediction and management of treatment response are missing.

ATC is by definition an advanced cancer [103]. Despite the Food and Drug Administration approval of the dual KI-treatment dabrafenib/trametinib for BRAF-mutated cases, the treatment approach of ATC is mainly palliative, based on a combination of surgery, chemotherapy, and radiotherapy [104]. Furthermore, due to tumor dedifferentiation, Tg is not secreted by the tumor and is not suitable for clinical management. Actually, no circulating biomarkers are available for ATC [105].

### 7.2. Role of cfDNA

The only study providing some insights about the role of cfDNA for the management of advanced thyroid cancers is the mentioned analysis by Allin et al. [52], which suggested that cfDNA analysis may be complementary and even empower the use of conventional biomarkers in patients with advanced thyroid malignancies.

Thirty-five DTC (17 PTC, 15 FTC, 3 PDTC), fifteen MTC, and one ATC were included. Overall, 45 out of 51 patients (88.2%) presented with advanced disease, the majority (43 subjects) harbouring distant metastases. The authors performed a matched tumor-plasma analysis with the NGS tissue evaluation followed by the droplet dPCR cfDNA assessment. The cfDNA analysis was performed serially, with a mean of five time points, thus allowing to accomplish a performance assessment and a comparison with conventional biomarkers.

For all tumor histotypes, authors found that modifications of cfDNA levels mirrored the evolution of disease status. This was observed for both the spontaneous tumor slope, with increased cfDNA concentrations preceding tumor progression, and the response to KI treatment, with decreasing and increasing trends being reported for responders and non-responders, respectively. More importantly, modifications of cfDNA concentrations as related to disease status changes were more accentuated and/or occurred earlier, as compared with conventional biomarkers.

Furthermore, cfDNA was detectable in two patients with Tg-negative follicular-derived cancer, one PTC with co-existing AbTg, and one ATC.

## 8. Conclusions

The main issue hampering the introduction of cfDNA testing into routine clinical practice of thyroid cancer, and more generally of all cancer patients, is the lack of standardization. Indeed, the studies performed to date were based on different detection methods, from qPCR to the novel dPCR and NGS, with variable accuracy. Furthermore, there are reproducibility and validity issues to be solved [106]. As a paradigm, the prevalence of cfBRAF^V600E^ in BRAF-mutated PTC, the most common cfDNA analysis performed in thyroid cancer, is widely variable between different series.

Despite these limitations, available data show high potential for cfDNA for improving thyroid cancer management (see Summary findings in Table 4 and visual reporting in Figure 1).

In the diagnostic setting, a remarkable capability for distinguishing thyroid cancer patients from benign subjects was shown by the pre-surgical assessment of cfBRAF^V600E^, cfDNA concentration/integrity index, and cfDNA methylation status. However, there is a need for studies focusing on indeterminate cytology, representing the real diagnostic challenge.

Concerning the prognostics, a relationship with the long-term outcome, specifically with poor survival, was found for the cfDNA RET M918T mutation in MTC and for the mutated cfPIK3CA in ATC. However, in PTC, there is robust evidence about the association between the cfBRAF^V600E^ and aggressive clinico-pathological features. Studies focusing on the association between cfDNA parameters and recurrence-free survival in DTC are needed.

Concerning the post-surgical phase (including the response to standard treatment, the follow-up, and the management of advanced disease), available studies are fewer and with less consistent results. However, there is emerging evidence about the ability of cfDNA concentrations in mirroring spontaneous and treatment-related disease slopes. This could have a breakthrough impact in the case of the unsuitability of conventional biomarkers.

**Figure 1 cancers-14-05370-f001:**
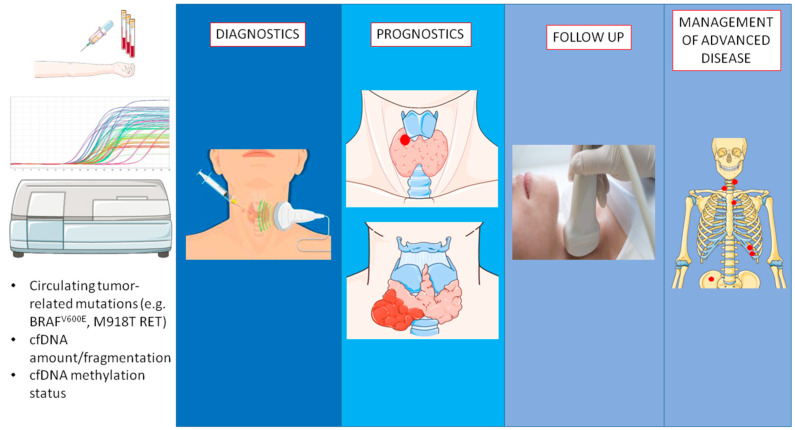
The diagnostic-therapeutic settings of thyroid cancer with the applicable cfDNA analyses. Parts of the figure were drawn by using pictures from Servier Medical Art (smart.servier.com (accessed on 27 October 2022). Servier Medical Art by Servier is licensed under a Creative Commons Attribution 3.0 Unported License (https://creativecommons.org/licenses/by/3.0/ (accessed on 27 October 2022).

In conclusion, cfDNA analysis has the potential to raise a concrete role in thyroid cancer management. However, due to the high costs and the need for skilled staff, the next research should perform a cost-benefit analysis aimed at defining the most useful clinical applications.

## Figures and Tables

**Table 1 cancers-14-05370-t001:** Design, methods, size, and findings of studies assessing the prevalence of cfBRAF^V600E^ in PTC patients. Only tumors with available BRAFV^600E^ mutational status were included. The somatic detection of BRAF^V600E^ was accomplished on primary tumors for all included cases. Prevalence data of cfBRAF^V600E^ were referred to the cfDNA evaluation as performed before thyroid removal.

Study (Year)	Design	Method/Matrix	N. of PTC	Prevalence of cfBRAF^V600E^ in BRAF Positive PTC (%)	Prevalence of cfBRAF^V600E^ in BRAF Negative PTC (%)
Kwak et al. (2013) [41]	prospective	qPCR/serum	94	0	0
Pupilli et al. (2013) [36]	prospective	allele-specific qPCR/plasma	22	91.7	30
Kim et al. (2015) [39]	retrospective	peptide nucleic acid clamp qPCR/plasma	72	6.1	not available
Condello et al. (2018) [42]	prospective	allele-specific qPCR + dPCR/plasma	46	0 *	0 *
Jensen et al. (2020) [49]	prospective	microfluidic dPCR preceded by COLD-PCR/plasma	57 **	42.1	not applicable
Patel et al. (2021) [37]	prospective	qPCR/plasma	20	33.3	40
Sato et al. (2021) [50]	prospective	droplet dPCR/plasma	22	31	0

cfBRAF^V600E^: circulating free BRAF^V600E^; qPCR: quantitative real-time PCR; dPCR: digital PCR; N.: Number. * data referred to both allele-specific qPCR and dPCR. ** only BRAF^V600E^-mutated PTC were included.

**Table 2 cancers-14-05370-t002:** Characteristics and findings of studies assessing the diagnostic role of cfDNA in thyroid cancer. Data about the accuracy of histological prediction were reported.

Study (Year)	Design	Study Population(N)	cfDNA Analysis *	Main Findings	Diagnostic Performance
Pupilli et al. (2013) [36]	prospective	38 PTC, 31 NG, 49 HC.	cfBRAF^V600E^	Higher percentage of mutated alleles in PTC, as compared with NG.	In the Thy3 cytology, 80% NPV and 33% PPV for PTC vs. NG.
Patel et al. (2021) [37]	prospective	68 PTC, 3 FTC, 38 NG.	cfBRAF^V600E^	All cfBRAF^V600E^ positive patients were affected with PTC classical variant.	(a) Specificity/PPV 100% for PTC vs. NG; (b) Sensitivity/NPV 22.1/41.8 for PTC vs. NG.
Zane et al. (2013) [34]	retrospective	86 PTC, 58 MTC, 9 ATC, 5 synchronous MTC-FTC, 23 FA, 19 HC.	cfDNA fragments of different length (low:ALU83, long:ALU244): absolute concentration.	Concentrations of both the high and the low-length amplicon were higher in thyroid cancer, as compared with HC.	(a) Low length amplicon: 0.91 AUC, 73.5% sensitivity, and 94.7% specificity for thyroid cancer vs. HC; (b) High length amplicon: 0.84 AUC, 67% sensitivity, and 100% specificity for thyroid cancer vs. HC.
Higazi et al. (2021) [44]	retrospective	18 PTC, 21 FTC, 21 MTC, 25 NG, 25 HC.	cfDNA fragments of different length (low:ALU83, long:ALU244): absolute concentration and integrity index **.	Concentrations of both the high and the low length amplicon and the integrity index were higher in thyroid cancer, as compared with NG and HC	(a) Integrity index: AUC 0.93, sensitivity 86%, and specificity 100% for thyroid cancer vs. HC; AUC 0.97, sensitivity 88%, and specificity 100% for thyroid cancer vs. NG; (b) ALU83 concentration: AUC 0.97, sensitivity 88%, and specificity 92% for thyroid cancer vs. HC; AUC 0.89, sensitivity 72%, and specificity 92% for thyroid cancer vs. NG; (c) ALU244 concentration: AUC 0.98, sensitivity 100%, and specificity 92% for thyroid cancer vs. HC; AUC 0.97, sensitivity 100%, and specificity 84% for thyroid cancer vs. NG; (d) in the Bethesda IV category specificity was 100% for the integrity index and 91% for ALU- 83 and -244
Klimaite et al. (2022) [43]	prospective	68 PTC and 31 NG, 86 HC.	cfDNA fragments of different length (low:β-actin99, long:β-actin394): absolute concentration and integrity index.	(a) Concentrations of both the high and the low length amplicon were higher in PTC, as compared with HC; (b) The integrity index was higher in PTC, as compared with both HC and NG.	(a) Integrity index: AUC 0.901, sensitivity 98.5%, and specificity 64% for PTC vs. HC; AUC 0.629, sensitivity 69.1%, and specificity 66.7% for PTC vs. NG; (b) β-actin99 concentration: AUC 0.593, sensitivity 75%, and specificity 73.3% for PTC vs. HC; (c) β-actin394 concentration: AUC 0.827, sensitivity 98.5%, and specificity 64% for PTC vs. HC.
Dutta et al. (2021) [69]	prospective	20 PTC, 4 FTC, 13 NG (all with indeterminate cytology [Bethesda III/IV])	Total cfDNA concentration	cfDNA concentration was higher in thyroid cancer, as compared with NG.	Sensitivity 100%, and specificity 92.3% for thyroid cancer vs. NG.
Hu et al. (2006) [38]	prospective	31 PTC, 7 FTC, 15 NG.	Methylation level of CALCA, CDH1, TIMP3, DAPK, and RARβ2 cfDNA	(a) Methylation of circulating TIMP3 and RARβ2 occurred solely in thyroid cancer; (b) Analysis of each of the included genes was able to predict malignancy with high specificity	(a) Sensitivity for thyroid cancer vs. NG: 29, 24, 21, 32, 32% for CALCA, CDH1, TIMP3, DAPK, RARβ2, respectively; Specificity for thyroid cancer vs. NG: 100, 100, 100, 95, 100% for CALCA, CDH1, TIMP3, DAPK, RARβ2, respectively: (b) The positivity of at least 1 gene showed 68% sensitivity and 95% specificity for thyroid cancer vs. NG

cfDNA: circulating free DNA; PTC: Papillary thyroid cancer; NG: Nodular goiter; HC: Healthy control; cfBRAF^V600E^: circulating free BRAF^V600E^; NPV: Negative predictive value; PPV: Positive predictive value; FTC: Follicular thyroid cancer; MTC: Medullary thyroid cancer; ATC: Anaplastic thyroid cancer; FA: Follicular adenoma; AUC: Area under the curve; APP: Amyloid Precursor Protein. * Parameters showing significant results were reported. ** integrity index is defined as the high/low length fragment ratio.

**Table 3 cancers-14-05370-t003:** Characteristics and findings of studies assessing the prognostic value of cfDNA in thyroid cancer.

Study (Year)	Design	Histology (N)	cfDNA Analysis	Clinico-Pathological Factors *	Outcome *
Kim et al. (2015) [39]	retrospective	BRAF^V600E^ positive PTC (49)	cfBRAF^V600E^	Lung metastases *p* < 0.001.	-
Jensen et al. (2020) [49]	prospective	BRAF^V600E^ positive PTC (57)	cfBRAF^V600E^	High tumor size *p* = 0.03; Gross extra-thyroidal extension *p* = 0.02; Pulmonary micro-metastases *p* = 0.04 High-risk ATA category *p* = 0.002	non-excellent treatment response *p* = 0.001
Sato et al. (2021) [50]	prospective	BRAF^V600E^ positive PTC (57)	cfBRAF^V600E^ **	Extra-thyroidal extension *p* = 0.01 High somatic BRAF^V600E^ fractional abundance *p* < 0.01	-
Patel et al. (2021) [37]	prospective	PTC (45)	cfBRAF^V600E^	High T-stage *p* < 0.05 Extra-thyroidal extension *p* < 0.05.	-
Cote et al. (2017) [51]	prospective	Sporadic RET M918T positive MTC (50) with persistent post-surgical disease	cfRET M918T ***.	Distant metastasis *p* = 0.03 Stage Ivc *p* = 0.01	Survival *p* < 0.0001.
Qin et al. (2021) [48]	retrospective	ATC (87)	Mutated cfPIK3CA ****	-	Survival *p* < 0.05
Ciampi et al. (2022) [47]	prospective	Sporadic MTC harboring somatic mutations (29)	cfDNA detection of the mutations identified on tumor tissue *****	High values of Ct and CEA *p* = 0.0307 and 0.0013, respectively ****** High somatic variation allele frequency *p* = 0.0468 *****	Persistent biochemical/structural disease *p* = 0.0005 *******
Zane et al. (2013) [34]	retrospective	PTC (86), MTC (58), ATC (9), synchronous MTC-FTC (5), FA(23)	cfDNA ALU83 and ALU244 concentrations and integrity index ********	Significant increase from FA/PTC to ATC (ALU83 concentration *p* < 0.0001; ALU244 concentration *p* < 0.0001; Integrity index *p* < 0.0001); Somatic BRAF^V600E^ mutation (Integrity index *p* = 0.02)	-
Klimaite et al. (2022) [43]	prospective	PTC (68)	cfDNA GADPH and β-actin99 concentrations; cfDNA β-actin394/99 integrity index	Tumor size > 2 cm (β-actin99 concentration *p* < 0.05; Integrity index β-actin394/99 *p* < 0.05)	-
Lan et al. (2020) [45]	retrospective	PTC (36)	cfDNA detection and cfDNA detection of somatic mutations ****	Distant metastasis (cfDNA detection *p* = 0.04; cfDNA detection of somatic mutations *p* = 0.015) Tumor size (cfDNA detection *p* = 0.001; cfDNA detection of somatic mutations *p* = 0.008) Invasiveness (cfDNA detection *p* = 0.01)	-
Dutta et al. (2021) [69]	prospective	PTC (33), FTC (4)	Total cfDNA concentration *********	Lymph node metastasis *p* = 0.005 Lymphovascular invasion *p* < 0.001 Capsular invasion *p* < 0.001 Extra-thyroidal extension *p* < 0.001 pTNM staging *p* = 0.005	-

N: Number; ATA: American Thyroid Association; CT: Calcitonin; CEA: Carcino-Embryonic Antigen; PTC: Papillary thyroid cancer; FTC: Follicular thyroid cancer; MTC: Medullary thyroid cancer; ATC: Anaplastic thyroid cancer; FA: Follicular adenoma; cfBRAF^V600E^: circulating free BRAF^V600E^; cfDNA: circulating free DNA; Tg: Thyroglobulin; cfPIK3CA: Circulating free Phosphatidylinositol-4,5-Bisphosphate 3-Kinase Catalytic Subunit Alpha; Glyceraldehyde 3-phosphate dehydrogenase. * Significant positive associations between the cfDNA test and poor clinico-pathological factors/unfavorable outcomes were reported. ** Samples with fractional abundance > 0.1% were considered positive. *** Samples with allelic fraction > 0.25% were considered positive. **** cfDNA analysis was performed by means of Next-Generation Sequencing techniques. ***** Samples with variation allele frequency ≥ 0.4% were considered positive. ****** referred to pre-operative ctDNA positivity. ******* referred to pre- and post- operative ctDNA positivity. ******** integrity index is defined as the high/low length fragment ratio. ********* determination of cfDNA concentration was performed by biospectrometer.

**Table 4 cancers-14-05370-t004:** Summary of findings.

Management Settings	N. of Studies (N. of Patients)	Main Evidence
Diagnostics	7 (812)	cfBRAF^V600E^, cfDNA concentration/fragmentation, cfDNA methylation status are useful tools for the presurgical identification of malignant nodules.
Prognostics	11 (632)	Circulating free tumor-related mutations are related to worst outcome/poor clinic-pathological features.
Follow-up	2 (212)	cfBRAF^V600E^ and cfDNA methylation status are useful tools for the detection of disease status during follow-up.
Management advanced disease	1 (51)	cfDNA concentration is a suitable marker of spontaneous and treatment-related morphological slope

N.: Number; cfBRAF^V600E^: circulating free BRAF^V600E^; cfDNA: circulating free DNA.

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
