# Peer review of "Cell-Free DNA Analysis within the Challenges of Thyroid Cancer Management"

_cancers, 2022, doi:10.3390/cancers14215370_

Round 1
Reviewer 1 Report
This study is well-written and collect together the news on cfDNA&thyroid cancers. Only minor remarks are needed before acceptance for publication: 1) although this is not a systematic review, at least the authors should report in a small AND dedicated paragraph the main search data (i.e., beginning and end search data; databases for consultation; inclusion and eclsusion criteria for the cited studies included in the tables). 2) the authors should create a new table of the summary findings so that the authors are easy to read and understand all the review, reporting practical messages for clinical/translation practice. 3) the authors could add 2 new references: one in the “diagnostic” section highlighting the limitation of nUS in children “Scappaticcio L, Maiorino MI, Iorio S, et al. Exploring the Performance of Ultrasound Risk Stratification Systems in Thyroid Nodules of Pediatric Patients. Cancers (Basel). 2021;13(21):5304”; one in the “follow-up” section “ Scappaticcio L, Trimboli P, Verburg FA, Giovanella L. Significance of "de novo" appearance of thyroglobulin antibodies in patients with differentiated thyroid cancer. Int J Biol Markers. 2020;35(3):41-49” highlighting the alert when using AbTg as a marker for DTC patients. 4) the authors should add one sentence at the end of the study why cfDNA analysis is emerging to date in clinical practice (i.e., few exsisting data; high costs; high expertise to apply in a thyroid team and etc).
Author Response
Reviewer: This study is well-written and collect together the news on cfDNA&thyroid cancers. Only minor remarks are needed before acceptance for publication.
Authors: We thank the author for appreciating the manuscript. We are really proud about that.
Reviewer: Although this is not a systematic review, at least the authors should report in a small AND dedicated paragraph the main search data (i.e., beginning and end search data; databases for consultation; inclusion and eclsusion criteria for the cited studies included in the tables).
Authors: We perfectly agree with the Reviewer and we thank him for such indication. Hence, we included a paragraph (Methods) fully dedicated to the search methods.
Reviewer: the authors should create a new table of the summary findings so that the authors are easy to read and understand all the review, reporting practical messages for clinical/translation practice.
Authors: We perfectly agree with the Reviewer and we thank him for such indication. Hence, we included a Table (Table 4) focused on the Sumary findings of the review.
Reviewer: The authors could add 2 new references: one in the “diagnostic” section highlighting the limitation of nUS in children “Scappaticcio L, Maiorino MI, Iorio S, et al. Exploring the Performance of Ultrasound Risk Stratification Systems in Thyroid Nodules of Pediatric Patients. Cancers (Basel). 2021;13(21):5304”; one in the “follow-up” section “ Scappaticcio L, Trimboli P, Verburg FA, Giovanella L. Significance of "de novo" appearance of thyroglobulin antibodies in patients with differentiated thyroid cancer. Int J Biol Markers. 2020;35(3):41-49” highlighting the alert when using AbTg as a marker for DTC patients.
Authors: As requested, we inserted the references indicated by the Reviewer.
Reviewer: The authors should add one sentence at the end of the study why cfDNA analysis is emerging to date in clinical practice (i.e., few exsisting data; high costs; high expertise to apply in a thyroid team and etc).
Authors: We perfectly agree with the Reviewer and we thank him for such indication. As requested, a dedicated sentence speculating about the concrete introduction of cfDNA analysis into the clinical practice of thyroid cancer management has been inserted at the end of the paragraph Conclusions.
Reviewer 2 Report
Marotta and colleagues propose a review of the literature based on the analysis of cfDNA in thyroid cancer and its usefulness for diagnostic and prognostic purposes. The Authors focused on the use of cfDNA analysis for cancer genomics, for diagnostic purpose, for prognostic stratification, and finally for therapeutic interventions, and for drug resistance detection.
The manuscript is well written and, though not totally original, cover all aspects of cfDNA analysis in thyroid cancer field with particular attention on the accuracy of methodologies used and on the application within the various clinical challenges.
Although the Authors summarize the results of several studies in a critically manner, I noted the lack of more recent literature on this topic (i.e. PMID: 33475693, 33601025, 33599171, 35152349). Moreover, they claimed to have covered all thyroid cancer histotypes but no data are reported about FTC (PMID: 29623111).
I suggest to include a paragraph describing the methodologies used with the relative advantages and disadvantages. I also propose to modify the figure 1 making it more complete, for example by including the different purposes of the cfDNA analysis.
Finally, I suggest to improve the manuscript adding a meta-analysis of the reported papers for which a similar study design has been applied.
Author Response
Reviewer: Marotta and colleagues propose a review of the literature based on the analysis of cfDNA in thyroid cancer and its usefulness for diagnostic and prognostic purposes. The Authors focused on the use of cfDNA analysis for cancer genomics, for diagnostic purpose, for prognostic stratification, and finally for therapeutic interventions, and for drug resistance detection. The manuscript is well written and, though not totally original, cover all aspects of cfDNA analysis in thyroid cancer field with particular attention on the accuracy of methodologies used and on the application within the various clinical challenges.
Authors: We thank the author for appreciating the manuscript. We are really proud about that.
Reviewer: Although the Authors summarize the results of several studies in a critically manner, I noted the lack of more recent literature on this topic (i.e. PMID: 33475693, 33601025, 33599171, 35152349). Moreover, they claimed to have covered all thyroid cancer histotypes but no data are reported about FTC (PMID: 29623111).
Authors: We greatly thank the Reviewer for giving us the chance of improving the review. Hence, we updated the manuscript with the suggested papers, with the exclusion of that about FTC (Song et al. 2018), focused on a case report. Indeed, as reported in the new paragraph Methods (requested by another Reviewer), studies with less than 20 patients were excluded. In order to report the most relevant data and to make the the review more concise (as request by another Reviewer), the Gouda paper about metastatic iodine-refractory DTC was cited only in the section (cfDNA analysis in thyroid cancers) about the detection methods, as not carrying relevant results.
Reviewer: I suggest to include a paragraph describing the methodologies used with the relative advantages and disadvantages.
Authors: We perfectly agree with the Reviewer and we thank him for such indication. Hence, we inserted in the revised version a part (see the section cfDNA analysis in thyroid cancers) dedicated to description, applications, advantages and disadvantages of the cfDNA analytical methods
Reviewer: I also propose to modify the figure 1 making it more complete, for example by including the different purposes of the cfDNA analysis.
Authors: We perfectly agree with the Reviewer and we thank him for such indication. Hence, we included a revised version of the Figure. This has been inserted in the Conclusions trying to provide a visual input of the different phases of thyroid cancer management with the applicable cfDNA analyses.
Reviewer 3 Report
This manuscript entitled “Cell-free DNA analysis within the challenges of thyroid cancer management” is a review paper. The manuscript is interesting and has clinical and scientific value. However, the manuscript itself is unnecessarily long, too long to be good. It is overloaded with so many data which are difficult to congest and understand. All sections contain so many citations (references) explained into details what all together does not provide a clear image of the main purpose of the study given by the title. All sections of the body text should be shortened, especially conclusions which have to be short and give the essentials to the readers.
Author Response
Reviewer: This manuscript entitled “Cell-free DNA analysis within the challenges of thyroid cancer management” is a review paper. The manuscript is interesting and has clinical and scientific value.
Authors: We thank the author for appreciating the manuscript. We are really proud about that.
Reviewer: However, the manuscript itself is unnecessarily long, too long to be good. It is overloaded with so many data which are difficult to congest and understand. All sections contain so many citations (references) explained into details what all together does not provide a clear image of the main purpose of the study given by the title. All sections of the body text should be shortened, especially conclusions which have to be short and give the essentials to the readers.
Authors: We thank the Reviewer for his suggestion. Following this indication, the revised version has been substantially shortened. Particularly, the paragraphs dedicated to the clinical challenges have been rendered more concise and those dedicated to the cfDNA analysis cleaned by less relevant data. For example, when focusing on diagnostics, both the Text and the corresponding Table (Table 2.) reported data about the correlation with histology, therefore excluding the cytology as analytical outcome. Similarly, when focusing on the follow-up phase, the anectodal and controversial data about the cfBRAFV600E modifications following surgery were removed. Ultimately, to highline the most robust evidence, we add Table 4, focusing on Summary findings.
Round 2
Reviewer 3 Report
Authors made all necessary corrections. The manuscript may be published.